# Involvement of Innate Immune System in the Pathogenesis of Sepsis-Associated Acute Kidney Injury

**DOI:** 10.3390/ijms241512465

**Published:** 2023-08-05

**Authors:** Takahiro Uchida, Muneharu Yamada, Dan Inoue, Tadasu Kojima, Noriko Yoshikawa, Shingo Suda, Hidenobu Kamohara, Takashi Oda

**Affiliations:** 1Department of Nephrology and Blood Purification, Kidney Disease Center, Tokyo Medical University Hachioji Medical Center, Tokyo 193-0998, Japan; 2Division of Critical Care and Emergency Medicine, Tokyo Medical University Hachioji Medical Center, Tokyo 193-0998, Japan

**Keywords:** acute kidney injury, fas ligand, natural killer, perforin, sepsis

## Abstract

Although experimental models have shown that the innate immune system is a main contributor to acute kidney injury (AKI), its involvement in human sepsis-associated AKI (SA-AKI) remains unclear. We retrospectively evaluated 19 patients with SA-AKI who were treated with continuous renal replacement therapy (CRRT). Serum cytokine, complement components, and the proportion and functions of innate immune cells, such as CD56^+^ T cells, CD56^+^ natural killer (NK) cells, and monocytes, were analyzed. There were no differences in the proportions of CD56^+^ T and NK cells between patients with SA-AKI and healthy controls. In patients with SA-AKI, fas ligand (FasL) expression in CD56^+^ T cells was significantly upregulated, and the proportion of perforin-positive CD56^+^ T cells tended to be higher than that in healthy controls. The positive rate of both FasL and perforin of CD56^+^ T cells was significantly higher than that of CD56^-^ T cells, which include cytotoxic T cells. Antigen-presenting capacity and phagocytic activity of monocytes in patients with SA-AKI were significantly decreased compared to those of healthy controls and did not recover soon after the initiation of CRRT. CD56^+^ T cells are involved in the disease processes of human SA-AKI through effector molecules such as FasL or perforin.

## 1. Introduction

Acute kidney injury (AKI) causes both high mortality and morbidity in affected patients. The development of sepsis-associated AKI (SA-AKI) is associated with higher mortality, despite recent improvements in the standard of care for sepsis [1,2]. The complicated pathogenesis of AKI is thought to contribute to the poor prognosis of SA-AKI; therefore, it is crucial to understand the pathogenesis of AKI more precisely. In addition to hemodynamic changes, cell apoptosis, cytokines or reactive oxygen species, and the immune system (including both innate as well as adaptive immune responses) have recently attracted much attention as important components of the pathogenesis of AKI [3,4].

Natural killer T (NKT) cells, which possess both the NK1.1 antigen and intermediate T cell receptor (TCR), and NK cells are representative innate immune cells in mice. It is well known that these cells exert critical antitumor and antimicrobial functions; however, they may attack normal cells upon inadequate overactivation, leading to shock and multiple organ failure (MOF) [5]. We have previously reported that mouse NKT cells are activated and cause AKI by directly injuring both renal tubular epithelial cells and vascular endothelial cells in response to various kinds of stimulation, such as cytokines or bacterial components. The TNF-α/fas ligand (FasL) system is mainly involved in the injury of tubular epithelial cells, whereas the perforin-mediated pathway is involved in vascular endothelial cell injury, and the two mechanisms play mutually independent roles [6]. Additionally, NK cells supplement the function of NKT cells during AKI development.

In humans, T cells possessing CD56 (CD56^+^ T cells), a cell surface marker of NK cells, have been considered a functional counterpart of mouse NKT cells; these cells are abundantly present in the liver, and most (70–80%) of them in the liver possess CD161, which is homologous to the NK1.1 antigen of mice [7,8]. Similar to NKT cells in mice, these cells exert strong antitumor effects and presumably damage normal cells when activated by cytokines or infectious pathogens, including lipopolysaccharide [9,10]. In line with this finding, we have previously shown that human-activated CD56^+^ T cells and CD56^bright^ NK cells (a subpopulation of CD56^+^ NK cells) severely damaged intrinsic renal cells such as tubular epithelial cells and glomerular endothelial cells in vitro, and we proposed that there are common pathogenic mechanisms between mouse and human conditions [6,11]. Thus, these innate immune cells could be a main effector in the pathogenesis of AKI; however, data showing their involvement in the disease process of human AKI are limited. Based on these findings, we investigated the roles of innate immune system components such as CD56^+^ T cells, NK cells, monocytes, and complements in SA-AKI in the present study.

## 2. Results and Discussion

Patient characteristics are summarized in Table 1.

Serum C3 and C4 levels were within normal ranges. In contrast, the serum levels of C5a and C5b-9 were significantly higher in patients with SA-AKI than those in healthy controls and did not decrease soon after the start of CRRT (Table 2).

Figure 1A shows flow cytometry analysis of PBMCs that were stained with anti-CD56 and anti-αβ TCR antibodies. No statistically significant difference was observed in the proportions of CD56^+^ T and NK cells between patients with SA-AKI and healthy controls (Figure 1B). FasL expression on CD56^+^ T cells of patients with SA-AKI was significantly upregulated compared to that in healthy controls but decreased soon after the start of CRRT (Figure 1C,D). FasL expression on CD56^+^ T cells was significantly upregulated compared to that on both NK and regular T cells (Figure 1E). The proportion of intracellular perforin-positive CD56^+^ T cells of patients with SA-AKI was numerically higher than that of healthy controls (SA-AKI group on day 1: 44.9 ± 11.1%; healthy controls: 26.9 ± 8.8%; Figure 1F). In patients with SA-AKI, the proportion of intracellular perforin-positive cells in both CD56^+^ T and NK cells was significantly higher than that in regular T cells at the time of initiation of CRRT (Figure 1G). The proportion of perforin-positive NK cells in patients with SA-AKI remained high, whereas an increase in the proportion of perforin-positive cells in CD56^+^ T cells was observed only transiently (Figure 1H).

The antigen-presenting capacity of monocytes (Figure 2A,B) and their phagocytic activity (Figure 2C,D) both significantly decreased in patients with SA-AKI compared to those of healthy controls and did not recover soon after the initiation of CRRT.

A study previously reported that both serum and urine FasL levels were increased in the early disease stage in an SA-AKI model [12]. In addition, we have previously shown that mouse NKT cells activated by their specific ligand or bacterial components caused AKI by damaging the renal tubular epithelial cells via the TNF-α/FasL pathway [13]. Consistent with these results, in the present study, FasL expression on CD56^+^ T cells, a functional counterpart of mouse NKT cells, was significantly upregulated in patients with SA-AKI compared to healthy controls. FasL expression on CD56^+^ T cells was also significantly upregulated than that on NK and CD56^-^ regular T cells, which include cytotoxic T cells at the time of CRRT initiation, suggesting their important role in disease pathogenesis.

We also showed a significantly increased proportion of perforin-positive cells both in CD56^+^ T and NK cells compared to regular T cells at the time of the introduction of CRRT, whereas the increase in the rate of perforin-positive cells in CD56^+^ T cells soon decreased. In the generalized Shwartzmann reaction, a model of MOF or septic shock, both NKT and NK cells in mice are suggested to play critical roles, supposedly using a perforin-mediated pathway [14]. In addition, we have shown that cytokine-stimulated human CD56^+^ T cells and a subpopulation of CD56^+^ NK cells injured intrinsic renal cells, such as tubular epithelial cells and glomerular endothelial cells, and that the cytotoxicity of CD56^+^ T cells toward glomerular endothelial cells was suppressed through inhibition of the perforin-mediated pathway by concanamycin A [6,11]. Both CD56^+^ T and NK cells produce and hold large amounts of cytotoxic effector molecules, including FasL and perforin [6], and perform antimicrobial functions that are indispensable for the control of sepsis; the proportion of perforin-positive cells reportedly increased after stimulation with bacterial agents, including lipopolysaccharide [10]. However inappropriately overactivated, these cells can even injure normal cells strongly, and the functions of these cells are considered two-edged swords. Therefore, it can be cautiously suggested that the inhibition of excessive activation of CD56^+^ T cells and/or NK cells is important for the treatment of SA-AKI and that targeting the FasL and perforin pathways is a promising approach, although we did not perform a transcriptomic analysis of these cells. In the present study, no significant difference was observed in the proportion of CD56^+^ T cells between patients with SA-AKI and healthy controls. The proliferation capacity of CD56^+^ T cells is reportedly weak [15]. Thus, it is possible that CD56^+^ T cells augment their cytotoxic functions without affecting their numbers.

Monocytes are known to recognize bacterial components and produce proinflammatory cytokines, including TNF-α, to activate NKT/NK cells [10]. In case of bacterial infection, it has also been suggested that monocytes cause renal injury through the production of proinflammatory cytokines and the activation of NKT/NK cells [13]. However, the functions of monocytes, such as phagocytic activity and antigen-presenting capacity, were decreased in the present study, suggesting immune paralysis [16]. Although the bactericidal activity of phagocytic cells is crucial for the treatment of both sepsis and SA-AKI, the immunoparalytic state of monocytes did not recover soon after the initiation of CRRT. It was recently reported that the expression of HLA-DR on monocytes in patients with severe sepsis was improved by polymyxin B hemoperfusion (PMX-HP) treatment [17]. However, the low proportion of patients receiving PMX-HP treatment in our study (15.8%) may have affected the results. Although the AN69 ST filter, which was reported to remove numerous cytokines by adsorption [18], was frequently used in the present study (68.4%), its effect on the innate immune system, especially monocytes’ functions, should be evaluated in future studies.

The complement system is a representative member of the innate immune system and plays an essential role in the defense against infections, including those leading to sepsis. However, its excessive activation in the later phases of sepsis could contribute to the development of MOF. It has been reported that the complement system is activated in AKI [4] and that C5b-9 deposition is observed in the tubulointerstitial area of an SA-AKI model [19]. Although the serum levels of C3 and C4 were normal, those of both C5a and C5b-9 were elevated in the patients in this study at the time of CRRT introduction. However, these levels did not decrease soon after the start of CRRT, suggesting the possibility of sustained inadequate overactivation of the complement system.

This study had some limitations. The first limitation was the small number of participants investigated, which could reduce the power of the study. Further studies with a larger number of patients are needed. Additionally, this was a retrospective study performed at a single center, despite the heterogeneous characteristics of the participants. The lack of a control group with SA-AKI who did not receive CRRT may make some of the results inconclusive. It would also be important to investigate whether patients with a milder form of SA-AKI show similar findings to those demonstrated in the present study or not.

Nevertheless, the present study suggests that innate immune cells, especially CD56^+^ T cells, are involved in the disease process of SA-AKI through effector molecules, such as FasL or perforin. Future studies should investigate whether targeting these cells and/or effector molecules is an effective therapeutic option for SA-AKI.

## 3. Materials and Methods

### 3.1. Patients and Their Clinical and Laboratory Data

Patients aged > 20 years who were diagnosed with SA-AKI and were treated with continuous renal replacement therapy (CRRT) between 2019 and 2020 at the Tokyo Medical University Hachioji Medical Center were retrospectively evaluated. Healthy volunteers who provided written informed consent were considered healthy controls (*n* = 10). This study was conducted according to the principles of the Declaration of Helsinki. The study protocol was approved by the Research Ethics Committee of Tokyo Medical University (Approval number: T2021-0049).

Clinical characteristics of the patients, including Sequential Organ Failure Assessment (SOFA) score, blood pressure, vasopressor use, information regarding isolated microorganisms, and routine laboratory data were retrieved from the medical records.

### 3.2. Measurements of Serum TNF-α, Circulating Immune Complexes (CIC), and Serum Complement Components

Serum C5a and C5b-9 levels were evaluated using ELISA kits (Quidel Corp., San Diego, CA, USA), essentially according to the manufacturer’s instructions [20]. Levels of serum TNF-α and CIC were measured using ELISA kits (R & D Systems, Inc., Minneapolis, MN, USA for TNF-α and Nissui Pharmaceutical Co., Ltd., Tokyo, Japan for CIC) by a clinical laboratory testing company (SRL, Inc., Tokyo, Japan).

### 3.3. Flow Cytometry

Peripheral blood mononuclear cells (PBMC) were collected from whole blood samples on the start date of CRRT (day 0), day 1, and day 2 using Lymphocyte Separation Medium (MP Biomedicals, Santa Ana, CA, USA). For the analysis of lymph cells, collected PBMCs were stained with PE-labeled anti-CD56 antibody (N901 (NKH-1), Beckman Coulter, Indianapolis, IN, USA) and PE-Cy5-labeled anti-αβTCR antibody (IP26A, Beckman Coulter). CD56^+^ T cells, NK cells, and regular T cells, which include cytotoxic T cells, were identified as CD56^+^ αβTCR^+^ cells, CD56^+^ αβTCR^-^ cells, and CD56^-^ αβTCR^+^ cells, respectively. To assess the effector molecules of these cells, the samples were stained with an FITC-labeled anti-perforin antibody (delta G9, Ancell Corp., Bayport, MN, USA) or anti-FasL antibody (SB93a, Thermo Fisher Scientific, Waltham, MA, USA). Cell membrane permeabilization was performed using a Fixation/Permeabilization Solution Kit (BD Biosciences, San Diego, CA, USA) immediately before incubation with an anti-perforin antibody. To assess the antigen-presenting capacity of monocytes, the cells were stained with PE-labeled anti-human leukocyte antigen (HLA)-DR antibody (MEM-12, Beckman Coulter). Flow cytometry analysis was performed using a CytoFLEX instrument (Beckman Coulter).

### 3.4. Evaluation of Phagocytic Activity

PBMCs were incubated with FITC-labeled 0.75 μm beads (Fluoresbrite YG Microspheres; Polysciences, Warrington, PA, USA) at 37 °C for 30 min [21]. The phagocytic activity of monocytes was assessed by evaluating the proportion of cells containing the FITC-labeled beads among the cells in the monocyte gate.

### 3.5. Statistical Analysis

Data are expressed as the mean ± SE. Differences between two experimental groups were assessed using Student’s t-test, and differences among more than three experimental groups were assessed using one-way ANOVA with post hoc analysis. *p*-values of <0.05 were considered statistically significant. All statistical analyses were performed using JMP software (version 14; SAS Institute Inc., Cary, NC, USA).

## Figures and Tables

**Figure 1 ijms-24-12465-f001:**
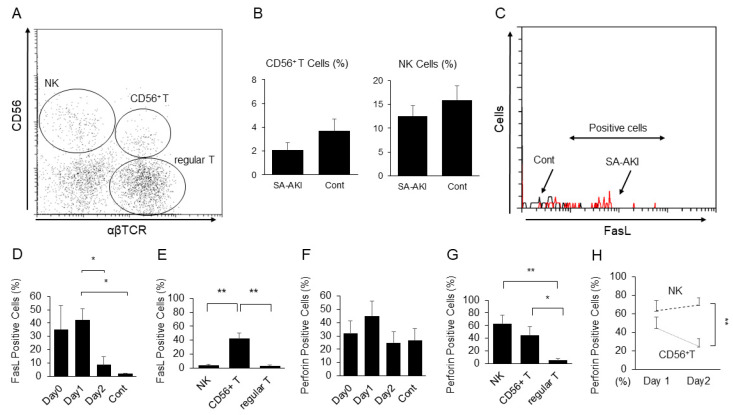
Flow cytometry analysis of peripheral blood mononuclear cells (PBMC) of patients with sepsis-associated acute kidney injury (SA-AKI) receiving continuous renal replacement therapy (CRRT). (**A**) PBMCs that were stained with anti-CD56 and anti-αβ T cell receptor antibodies. (**B**) The proportions of CD56^+^ T cells (left) and CD56^+^ natural killer (NK) cells (right) of patients with SA-AKI and healthy controls (Cont). (**C**) Representative histogram showing fas ligand (FasL) expression on CD56^+^ T cells of a patient with SA-AKI (red) and a healthy control (black). (**D**) Change in the proportions of FasL-expressing CD56^+^ T cells after the start of CRRT. (**E**) The proportion of FasL-positive lymph cells on day 1. (**F**) Change in the proportions of intracellular perforin-positive CD56^+^ T cells after the start of CRRT. (**G**) The proportion of perforin-positive lymph cells on day 1. (**H**) Change in the proportions of perforin-positive cells in CD56^+^ T cells and NK cells after the start of CRRT. N.S.: not significant. * *p* < 0.05; ** *p* < 0.01.

**Figure 2 ijms-24-12465-f002:**
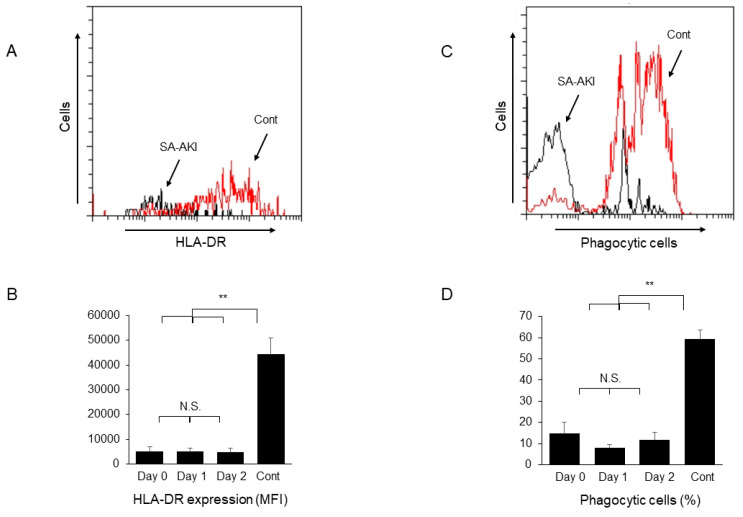
Decreased antigen-presenting capacity and phagocytic activity of peripheral blood monocytes of patients with sepsis-associated acute kidney injury (SA-AKI). (**A**) Representative histogram showing human leukocyte antigen (HLA)-DR expression on monocytes of a patient with SA-AKI (black) and a healthy control (red). (**B**) Change in HLA-DR expression (mean fluorescence intensity; MFI) on monocytes after the start of continuous renal replacement therapy (CRRT). (**C**) Representative histogram showing monocytes containing FITC beads of a patient with SA-AKI and a healthy control. (**D**) Change in the percentage of phagocytic cells in monocytes after the start of CRRT. Cont: healthy controls, N.S.: not significant. ** *p* < 0.01.

**Table 1 ijms-24-12465-t001:** Characteristics of the patients.

	Data
Number	19
Age (years)	71.3 ± 2.2
Sex (M/F)	12/7
Serum creatinine at enrollment (mg/dL)	3.66 ± 0.40
SOFA score	12.8 ± 0.8
Mean blood pressure (mmHg)	64.0 ± 2.1
Vasopressor use	100%
Isolated organism (GPC/GNR/other)	5/15/0
Site of infection (GI/UTI/lung/skin)	10/4/2/1
AN69 ST membrane use	68.4%
Polymyxin B hemoperfusion treatment	15.8%
CRP (mg/dL)	18.2 ± 2.1
Procalcitonin (ng/mL)	74.4 ± 21.0
TNF-α (pg/mL)	22.3 ± 5.4
CIC-mRF (≧4.2 μg/mL)	40%
C3 (mg/dL)	81.6 ± 10.1
C4 (mg/dL)	19.6 ± 1.8
C5a (ng/mL)	24.1 ± 4.5
C5b-9 (ng/mL)	2201 ± 678

Data are presented as mean ± SEM or number. CIC-mRF, circulating immune complex assessed by monoclonal rheumatoid factor assay; SOFA, Sequential Organ Failure Assessment.

**Table 2 ijms-24-12465-t002:** Serum C5a and C5b-9 levels after the start of continuous renal replacement therapy.

	Healthy Controls	Day 0	Day 1	Day 2
C5a (ng/mL)	13.2 ± 1.4 *	24.1 ± 4.5	20.6 ± 7.5	22.8 ± 6.1
C5b-9 (ng/mL)	1046 ± 181 *	2201 ± 678	2326 ± 1077	2214 ± 960

Data are presented as mean ± SEM. * *p* < 0.05.

## Data Availability

All datasets generated for this study are included in the article. Further inquiries can be directed to the corresponding author.

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
