# Peer review of "Involvement of Innate Immune System in the Pathogenesis of Sepsis-Associated Acute Kidney Injury"

_ijms, 2023, doi:10.3390/ijms241512465_

Round 1

Reviewer 1 Report

Well written it would be of importance if you could perform transcriptomic analysis on your cells to further prove your hypothesis. In addition the paper would be of greater impact if you could demonstrate this on milder forms of AKI.

This paper expands the field of knowledge regarding the pathophysiology of disease once thought to be simply ischemic in nature.  There has been increasing evidence that immunity and inflammation are possibly related to cytokines and activation of innate immunity.  this study adds mounting evidence that innate immunity by upregulation via Fasl ligand and perforin terminal compliment   of innate immune effectors in patients with severe SA AKI. (those pts on crrt) The conclusions are supported by their findings.  However it would be of great importance if the same "immunologic synapse occurs in milder forms of SA AKI.  This is a solid paper and expands on previous work.  This study should be reproduced in larger number of patients.  It is of importance that what the authors demonstrate maybe a protective measure in inducing apoptosis at least in regards to FASl.  

The methodology of paper is adequate perhaps they could demonstrate apoptosis in their cell populations such as monocytes this would be difficult.  What occurs to controls if lps is added ?  This should not limit publicationOf interest in further studies transcriptomic should be employed to observe if there are different endotypes.

The paper is highly referenced and appropriate.

I mentioned earlier the paper should be accepted. it would be a better study if they could do some of these other experiments in particular addition of lps to controls in order to demonstrate this is lps mediated however it should not preclude publication.

Author Response

We sincerely thank you for your kind and constructive comments and suggestions regarding our manuscript. Our point-by-point responses are shown below.

Well written it would be of importance if you could perform transcriptomic analysis on your cells to further prove your hypothesis. In addition the paper would be of greater impact if you could demonstrate this on milder forms of AKI.

This paper expands the field of knowledge regarding the pathophysiology of disease once thought to be simply ischemic in nature.  There has been increasing evidence that immunity and inflammation are possibly related to cytokines and activation of innate immunity.  this study adds mounting evidence that innate immunity by upregulation via Fasl ligand and perforin terminal compliment   of innate immune effectors in patients with severe SA AKI. (those pts on crrt) The conclusions are supported by their findings.  However it would be of great importance if the same "immunologic synapse occurs in milder forms of SA AKI.  This is a solid paper and expands on previous work.  This study should be reproduced in larger number of patients.  It is of importance that what the authors demonstrate maybe a protective measure in inducing apoptosis at least in regards to FASl. 

The methodology of paper is adequate perhaps they could demonstrate apoptosis in their cell populations such as monocytes this would be difficult.  What occurs to controls if lps is added ?  This should not limit publicationOf interest in further studies transcriptomic should be employed to observe if there are different endotypes.

The paper is highly referenced and appropriate.

I mentioned earlier the paper should be accepted. it would be a better study if they could do some of these other experiments in particular addition of lps to controls in order to demonstrate this is lps mediated however it should not preclude publication.

Response:

Thank you for the comments.

First, although we did not perform transcriptomic analysis in this study, we totally agree that such analysis would further confirm our hypothesis. Therefore, we have added some comments on this matter in the revised manuscript (lines 198 - 201).

It is indeed very important to investigate whether patients with milder form of SA-AKI show the similar findings to those demonstrated in the present study or not. We also added some comments on this matter in the revised manuscript (lines 235 - 237).

We completely agree that the small number of participants was indeed the limitation of the present study, and that further studies with larger number of patients are needed. Therefore, we clearly stated this point in the revised manuscript (lines 231 - 232).

Finally, you recommended to perform experiments on the stimulation of PBMCs of healthy controls by LPS. Actually, there was a report on the indicated experiments, which demonstrated that the proportion of perforin-positive cells were increased after stimulation with bacterial agents, including LPS (Ref. 10 in the original manuscript). We have therefore added the detailed discussion on this matter in the revised manuscript (lines 60 and 194 - 196).

Reviewer 2 Report

Dear Colleagues,

I want to congratulate you on your work.

The article properly combines fundamental research with a clinical study, achieving the desired goal.

Capturing the peculiarities of the immune system response in the pathogenesis of sepsis-associated acute kidney injury is essential for finding new therapeutic targets.

I agree with the publication of the work in its existing form.

Kind Regards,

Upadte:

I have read this manuscript with much attention and interest.

It is an original study conducted on human subjects that expands knowledge regarding the mechanisms of kidney damage from sepsis.

The study presents interesting results, which in the future may have an impact on the therapeutic approach to renal injury in patients with sepsis.

The manuscript has many strong points, among which I mention the following:

The introduction is synthetic and pragmatic. It mentions the most important landmarks in the literature, emphasizing the existence of studies on animal models and the paucity of information from studies with human subjects. The authors emphasize the need to deepen knowledge and the gaps that the current study aims to fill. The aim and the scope of the current study are clearly presented.

The Material and Method section is well structured and provides a balanced amount of technical data, which gives robustness to the results obtained.

The Results section is the strength of the manuscript. In addition to the scientific value of the obtained data, the clarity of its communication should be important. The visual component was given great importance, with the graphs and tables of systematization and synthesis giving impact to the results obtained.

The discussion section is an integration of the results obtained from the already existing literature. Logical bridges are created between notions, data from preclinical studies are correlated with clinical ones. The debate is correct, rational, anchored in concrete data and outlines the premises of further research.
Considering all these positive elements, I recommend the publication of this manuscript.

Author Response

We sincerely thank you for your detailed and positive comments on our manuscript.